# Understanding R1-Zero-Like Training: A Critical Perspective

**Zichen Liu**[*†1,2], **Changyu Chen**[*1,3], **Wenjun Li**[*3], **Penghui Qi**[*1,2],
**Tianyu Pang**[1], **Chao Du**[1], **Wee Sun Lee**[2], **Min Lin**[1]
[1]Sea AI Lab
[2]National University of Singapore
[3]Singapore Management University

## Abstract

DeepSeek-R1-Zero has shown that reinforcement learning (RL) at scale can directly enhance the reasoning capabilities of LLMs without supervised fine-tuning. In this work, we critically examine R1-Zero-like training by analyzing its two core components: *base models* and *RL*. We investigate a wide range of base models, including DeepSeek-V3-Base, to understand how pretraining characteristics influence RL performance. Our analysis reveals that **DeepSeek-V3-Base already exhibit "Aha moment"**, while **Qwen2.5 base models demonstrate strong reasoning capabilities even without prompt templates**, suggesting potential pretraining biases. Additionally, we identify an optimization bias in Group Relative Policy Optimization (GRPO), which artificially increases response length (especially for incorrect outputs) during training. To address this, we introduce **Dr. GRPO**, an unbiased optimization method that improves token efficiency while maintaining reasoning performance. Leveraging these insights, we present a minimalist R1-Zero recipe that achieves 43.3% accuracy on AIME 2024 with a 7B base model, establishing a new state-of-the-art.

 **https://github.com/sail-sg/understand-r1-zero**[1]

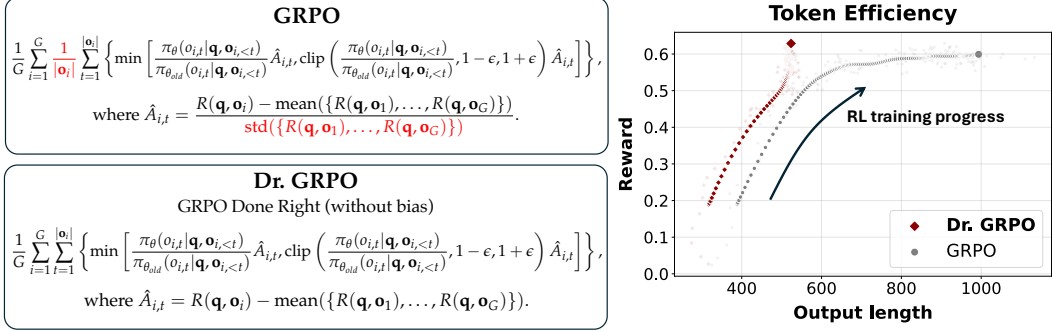

Figure 1: **Left**: Dr. GRPO introduces simple yet significant modifications to address the biases in GRPO (Shao et al., 2024), by removing the length and std normalization terms. **Right**: Our unbiased optimizer effectively prevents the model from generating progressively longer incorrect responses, thereby enhancing token efficiency.

## 1 Introduction

DeepSeek-R1-Zero (Guo et al., 2025) revolutionizes the pipeline of large language model (LLM) post-training by introducing the *R1-Zero-like training paradigm*: directly applying RL to base LLMs without relying on supervised fine-tuning (SFT) as a preliminary step. This new paradigm is appealing due to its simplicity and the demonstrated **RL scaling phenomenon**: the model reasoning capabilities improve along with a continual increase in

---

[*]Core Contributors.
[†]Project Lead.
[1]Developed with the LLM RL framework Oat: https://github.com/sail-sg/oat.

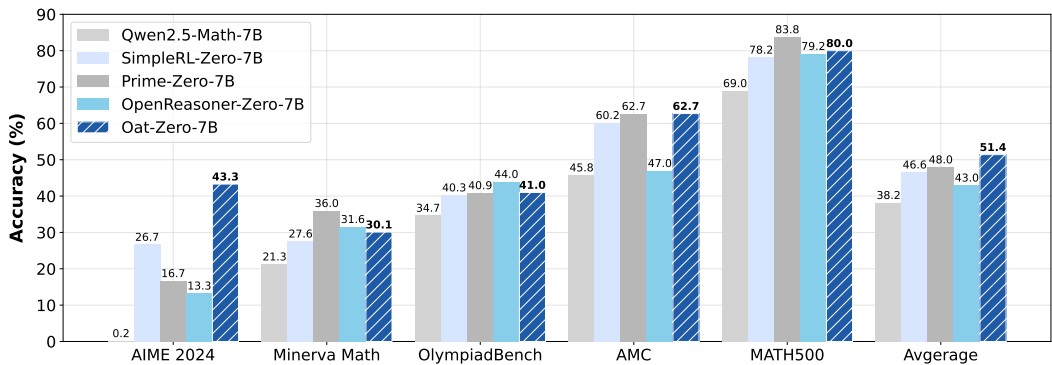

Figure 2: Model performance comparison. **Oat-Zero-7B** is RL-tuned with our minimalist recipe described in Sec. 1 (third paragraph). Please see App. B for more results.

model's response length. This phenomenon is also accompanied by the "Aha moment", at which the model learns emergent skills such as self-reflections.

In this paper, we aim to understand R1-Zero-like training by studying two essential components: *base models* and *RL*. In the first part, we investigate various attributes of base models, with the focus on the **Qwen2.5** model family (Yang et al., 2024a;b), which has been used in recent attempts to reproduce R1-Zero (Pan et al., 2025; Zeng et al., 2025; Liu et al., 2025b; Hu et al., 2025), as well as **DeepSeek-V3-Base** (Liu et al., 2024), from which the real R1-Zero model was RL-tuned. In the second part, we identify the **bias in optimization of GRPO** (Shao et al., 2024), which may lead to progressively longer *incorrect* responses. To this end, we propose a simple modification to eliminate the bias, i.e., to get GRPO Done Right (**Dr. GRPO**), which leads to **better token efficiency** (highlighted in Fig. 1).

Our analysis on base models and RL suggests a **minimalist recipe** for R1-Zero-like training: we RL-tune Qwen2.5-Math-7B using the (unbiased) Dr. GRPO algorithm on MATH (Hendrycks et al., 2021) level 3-5 questions with the Qwen-Math template, and achieve state-of-the-art performance (Fig. 2) with only 27 hours compute on $8\times$ A100 GPUs. We hope our findings presented in this paper, models released, and the codebase open-sourced could benefit future research in the field. As an overview, we summarize the takeaways of this paper below:

---

**Overview of takeaways**

- (Sec. 2.1) Template is crucial to make base models **answer questions** instead of completing sentences. In addition, all base models already possess math-solving capability prior to RL.

- (Sec. 2.2) Intriguingly, Qwen-2.5 base models get an **immediate** $\sim 60\%$ **improvement by not using template**, making us hypothesize that they may pretrain on concatenated question-answer texts when cooking the models.

- (Sec. 2.3) Nearly all base models already exhibit the "Aha moment", **including DeepSeek-V3-Base**.

- (Sec. 3.1, Sec. 3.2) Dr. GRPO effectively fixes GRPO's bias in optimization, achieving **better token efficiency**.

- (Sec. 3.3) Model-template **mismatch** can destroy reasoning capabilities before RL reconstructs it.

- (Sec. 3.4) **Math pretraining on Llama-3.2-3B** improves its RL ceiling.

---

## 2 Analysis on Base Models

In this section, we scrutinize a wide range of base models, including the Qwen-2.5 family (Yang et al., 2024a;b), Llama-3.1 (Grattafiori et al., 2024) and DeepSeek series (Liu et al., 2024; Shao et al., 2024; Guo et al., 2025), asking them 500 questions sampled from the MATH (Hendrycks et al., 2021) training set and analyzing their responses.

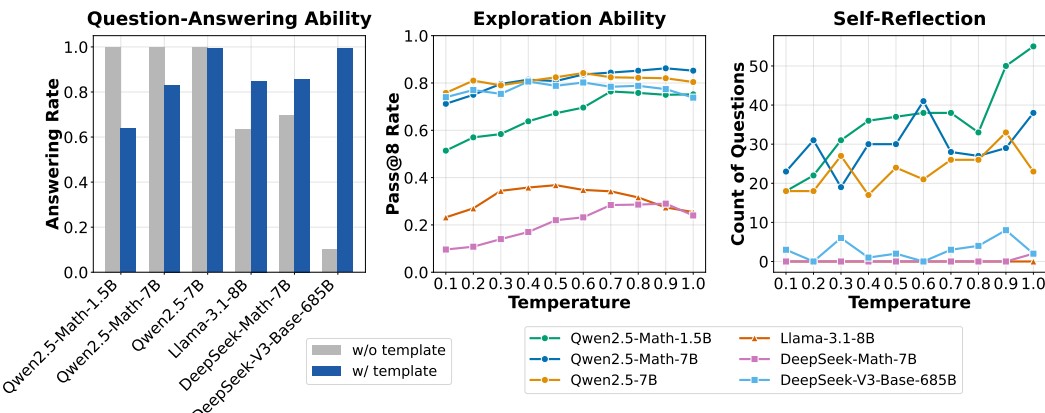

Figure 3: Model attributes across three aspects. **Question-Answering Ability**: the extent to which a pretrained language model provides a direct answer to a question rather than continuing or expanding upon it; **Exploration Ability**: pass@8 measures how well base models explore; **Self-Reflection**: counts are obtained through cross-validation between keyword-based detection and LLM-based detection, as detailed in Appendix D.

## 2.1 R1-Zero Trainability: Templates Construct Exploratory Base Policies

Since training from a base model is a fundamental setting of the R1-Zero-like paradigm, we first investigate whether widely used open-source base models, which are typically trained for sentence completion (i.e., $p_\theta(\mathbf{x})$), can have their question-answering capabilities effectively elicited through appropriate templates, thereby functioning as a question-answering base policy $\pi_\theta(\cdot|\mathbf{q})$. In addition to the *R1 template* (Template 1) in Guo et al. (2025), we consider the *Qwen-Math template* (Template 2) used by Zeng et al. (2025), as well as *No template* (Template 3):

---

**Template 1 (*R1 template*).** A conversation between User and Assistant. The User asks a question, and the Assistant solves it. The Assistant first thinks about the reasoning process in the mind and then provides the User with the answer. The reasoning process is enclosed within <think> </think> and answer is enclosed within <answer> </answer> tags, respectively, i.e., <think> reasoning process here </think> <answer> answer here </answer>.\nUser: {question}\nAssistant: <think>

**Template 2 (*Qwen-Math template*).** <|im_start|>system\nPlease reason step by step, and put your final answer within \\boxed{}.<|im_end|>\n<|im_start |>user\n{question}<|im_end|>\n<|im_start|>assistant\n

**Template 3 (*No template*).** {question}

---

**Experimental settings**. We include Qwen2.5-Math-1.5B, Qwen2.5-Math-7B, Qwen2.5-7B, Llama-3.1-8B, DeepSeek-Math-7B and DeepSeek-V3-Base-685B for experiments. For each model, we first apply *No template* to get the model responses, then let GPT-4o-mini to judge whether the model responses are in an answering format (regardless of quality) or in a sentence-completion pattern. We record the percentage of responses that tend to answer the question as the metric. We then apply both *R1 template* and *Qwen-Math template* to obtain model responses, and determine the most suitable template for each model based on the metric. Finally, we evaluate the pass@8 accuracy of each model with the corresponding template to assess whether the base policies can explore rewarding trajectories for RL improvement.

**Results**. The left plot of Fig. 3 shows how well base models (with or without templates) answer the provided questions. We observe that Llama and DeepSeek models all improve the answering ability by employing the proper template (R1 template). However, Qwen2.5 models work best (with 100% answering rate) when no template is used. This intriguing property motivates further investigation which will be discussed in Sec. 2.2. Meanwhile, the lowest answering rate with no template suggests that DeepSeek-V3-Base is a nearly pure base model. This observation motivates us to explore whether a pure base model

| Base model + Template | AIME24 | AMC | MATH500 | Minerva | OlympiadBench | Avg. |
|---|---|---|---|---|---|---|
| Qwen2.5-Math-1.5B | | | | | | |
| (4-shot prompting) | 0.0 | 20.0 | 50.4 | 12.1 | 15.9 | 19.7 |
| R1 template | 0.0 | 9.6 | 21.2 | 6.6 | 2.2 | 7.9 |
| Qwen template | **20.0** | 32.5 | 33.0 | 12.5 | 22.8 | 24.2 |
| No template | 16.7 | **43.4** | **61.8** | **15.1** | **28.4** | **33.1** |
| Qwen2.5-Math-7B | | | | | | |
| (4-shot prompting) | 3.3 | 22.5 | 61.6 | 10.7 | 20.9 | 23.8 |
| R1 template | 0.0 | 0.0 | 0.0 | 0.0 | 0.1 | 0.0 |
| Qwen template | **16.7** | 38.6 | 50.6 | 9.9 | 16.6 | 26.5 |
| No template | 0.2 | **45.8** | **69.0** | **21.3** | **34.7** | **38.2** |

Table 1: Qwen2.5-Math models might be pretrained on concatenated question-answer text, resulting in peak performance when **no template** is applied.

like DeepSeek-V3-Base demonstrates the Aha moment (Sec. 2.3). The middle plot of Fig. 3 shows the pass@8 accuracy of different base models (with template) at different sampling temperatures. This metric can serve as an indicator of base policy's exploration ability. For example, if a base policy cannot even sample a single trajectory that leads to the correct final answer, it is impossible for RL to improve the policy because there is no reward signal. Our results demonstrate that all tested models are exploratory (thus ready for RL), with Qwen2.5 models performing the best (even surpassing DeekSeek-V3-Base). This might partially explain that most R1-Zero projects (Zeng et al., 2025; Hu et al., 2025) are based on Qwen2.5 models.

## 2.2 Qwen-2.5 Models Unlock the Best Performance When Discarding Template

We next dig into the intriguing observation (c.f. Fig. 3(Left)) that all Qwen2.5 base models readily serve as chat models even without any template. We take a step further to evaluate the reasoning ability of Qwen2.5-Math models on five standard benchmarks: AIME 2024 (Li et al., 2024a), AMC (Li et al., 2024a), MATH500 (Hendrycks et al., 2021), Minerva Math (Lewkowycz et al., 2022), and OlympiadBench (He et al., 2024). Following common practice, we use greedy decoding and limit the sampling budget to 3000 tokens.

As shown in Table 1, not using any template can drastically boost the average performance, resulting in an improvement of about 60% compared to the traditional 4-shot prompting. Since Qwen2.5-Math (Yang et al., 2024b) uses chat model's data (question-answer pairs) during the pretraining stage, we hypothesize that they might pretrain on the concatenated text to maximize $\log p_\theta(\mathbf{q}; \mathbf{o})$ directly. If our hypothesis turns out true, we shall be more careful about using Qwen2.5 models to reproduce DeepSeek-R1-Zero, since the base models are already SFT-like without templates.

## 2.3 Aha Moment Already Appears in Base Models Including DeepSeek-V3-Base

One of the most inspiring results of DeepSeek-R1-Zero is the emergence of self-reflection behaviors, a.k.a., Aha moment, through pure RL training. A few prior studies (Liu et al., 2025b; Yeo et al., 2025) have suggested that there may not be Aha moment in open-source R1 replications because the base models they use already exhibit self-reflection keywords. However, they have not tested DeepSeek-V3-Base, on which the real R1-Zero model was RL-tuned. We complete this missing piece by hosting DeepSeek-V3-Base-685B ourselves and investigating its responses to the 500 MATH questions with the R1 template. From the right plot of Fig. 3, we can observe that DeepSeek-V3-Base also generates a decent amount of self-reflections, further validating the claims of Liu et al. (2025b). We also show examples in App. E (Fig. 13) where DeepSeek-V3-Base generates keywords such as "Aha" and "wait".

An additional important question is whether self-reflection behaviors are associated with improved model performance after RL training. To investigate this, we host DeepSeek-R1-Zero and analyze its responses to the same questions from the MATH dataset. Although self-reflection behaviors occur more frequently in R1-Zero, we observe that these behaviors are not positively correlated with higher accuracy. Detailed analysis can be found in App. F.

## 3 Analysis on Reinforcement Learning

Language model generation can be formulated as a token-level Markov Decision Process (MDP) $\mathcal{M} = (\mathcal{S}, \mathcal{A}, r, p_{\mathcal{Q}})$. At each generation step $t$, the state $s_t \in \mathcal{S}$ is the concatenation of the input question and the output response generated so far: $s_t = \mathbf{q}; \mathbf{o}_{<t} = [q_1, \ldots, q_M, o_1, \ldots, o_{t-1}]$. The policy $\pi_\theta(\cdot|s_t)$ will select the next token $o_t$ from the vocabulary $\mathcal{A}$, resulting in a deterministic transition to the next state $s_{t+1} = s_t; [o_t]$. The generation process starts from sampling an initial state $s_1 = \mathbf{q} \sim p_{\mathcal{Q}}$ from a set of questions, and stops when the autoregressive policy generates the [eos] token or exhausts the budget.

Typically, we maximize the entropy-regularized objective (Schulman et al., 2017a):

$$\mathcal{J}(\pi_\theta) = \underset{\mathbf{q} \sim p_{\mathcal{Q}}}{\mathbb{E}} \left[ \underset{\mathbf{o} \sim \pi_\theta(\cdot|\mathbf{q})}{\mathbb{E}} [R(\mathbf{q}, \mathbf{o})] - \beta \mathbb{D}_{KL}[\pi_\theta(\cdot|\mathbf{q})) || \pi_{\text{ref}}(\cdot|\mathbf{q})] \right], \tag{1}$$

where $R(\mathbf{q}, \mathbf{o}) = \sum_{t=1}^{|\mathbf{o}|} r(s_t, o_t)$ is the return (Sutton & Barto, 2018) of the trajectory $\mathbf{q}; \mathbf{o}$, and $\pi_{\text{ref}}$ is a reference policy. The KL regularization term is usually adopted ($\beta > 0$) for reinforcement learning from human feedback (Christiano et al., 2017), where $r$ is a **reward model** learned from data collected by $\pi_{\text{ref}}$. In this case, regularization helps prevent $\pi_\theta$ from deviating too far from the distribution where the reward model is accurate (Jaques et al., 2019; Stiennon et al., 2020). However, RL-tuning reasoning models typically employs **rule-based verifiers** as $r$ (Lambert et al., 2024), eliminating the concerns of distributional shift. This allows us to remove the KL term, which not only saves the memory and computation required by $\pi_{\text{ref}}$ during training, but also potentially leads to better performance for R1-Zero-like training (Hu et al., 2025). We will assume $\beta = 0$ throughout this paper.

**Policy optimization algorithms**. To optimize $\pi_\theta$ with the above objective (Eq. (1) with $\beta = 0$), Proximal Policy Optimization (PPO) (Schulman et al., 2017b) maximizes the following surrogate objective:

$$\mathcal{J}_{PPO}(\pi_\theta) = \mathbb{E}_{\mathbf{q} \sim p_{\mathcal{Q}}, \mathbf{o} \sim \pi_{\theta_{\text{old}}}(\cdot|\mathbf{q})}$$
$$\sum_{t=1}^{|\mathbf{o}|} \left\{ \min \left[ \frac{\pi_\theta(o_t|\mathbf{q}, \mathbf{o}_{<t})}{\pi_{\theta_{\text{old}}}(o_t|\mathbf{q}, \mathbf{o}_{<t})} \hat{A}_t, \text{clip}(\frac{\pi_\theta(o_t|\mathbf{q}, \mathbf{o}_{<t})}{\pi_{\theta_{\text{old}}}(o_t|\mathbf{q}, \mathbf{o}_{<t})}, 1 - \epsilon, 1 + \epsilon) \hat{A}_t \right] \right\}, \tag{2}$$

where $\pi_{\theta_{\text{old}}}$ is the policy before the update, $\epsilon$ is the clipping hyperparameter, and $\hat{A}_t$ is an estimator of the advantage function of the $t$-th token. A standard way to estimate $\hat{A}_t$ is to compute the Generalized Advantage Estimation (GAE) (Schulman et al., 2015) with a learned value model $V_\phi$. However, in the context of LLM RL-tuning, learning the value model is computationally expensive, so methods that estimate $\hat{A}_t$ without $V_\phi$ are practically preferred. For example, Shao et al. (2024) proposed GRPO, which first samples a group of responses $\{\mathbf{o}_1, \ldots, \mathbf{o}_G\}$ per question and computes their returns $\mathbf{R} = \{R_1, \ldots, R_G\}$, then sets the advantage of all tokens from $\mathbf{o}_i$ as $\hat{A}_t = \frac{R_i - \text{mean}(\mathbf{R})}{\text{std}(\mathbf{R})}$.

### 3.1 GRPO Leads to Biased Optimization

In Deepseek-R1-Zero (Guo et al., 2025), a notable trend is the consistent increase in response length throughout the training process. This is frequently interpreted as an indication of the development of advanced reasoning abilities such as self-reflection. Recent studies (Pan et al., 2025; Zeng et al., 2025; Hu et al., 2025) have replicated this phenomenon using various algorithms and implementations. However, we argue that **the observed increase in response length may also be attributed to a bias inherent in the GRPO (Shao et al., 2024) objective function**:

$$\mathcal{J}_{GRPO}(\pi_\theta) = \mathbb{E}_{\mathbf{q} \sim p_{\mathcal{Q}}, \{\mathbf{o}_i\}_{i=1}^G \sim \pi_{\theta_{old}}(\cdot|\mathbf{q})}$$
$$\frac{1}{G} \sum_{i=1}^G \frac{1}{|\mathbf{o}_i|} \sum_{t=1}^{|\mathbf{o}_i|} \left\{ \min \left[ \frac{\pi_\theta(o_{i,t}|\mathbf{q}, \mathbf{o}_{i,<t})}{\pi_{\theta_{old}}(o_{i,t}|\mathbf{q}, \mathbf{o}_{i,<t})} \hat{A}_{i,t}, \text{clip} \left( \frac{\pi_\theta(o_{i,t}|\mathbf{q}, \mathbf{o}_{i,<t})}{\pi_{\theta_{old}}(o_{i,t}|\mathbf{q}, \mathbf{o}_{i,<t})}, 1 - \epsilon, 1 + \epsilon \right) \hat{A}_{i,t} \right] \right\}, \tag{3}$$

where

$$\hat{A}_{i,t} = \frac{R(\mathbf{q}, \mathbf{o}_i) - \text{mean}(\{R(\mathbf{q}, \mathbf{o}_1), \ldots, R(\mathbf{q}, \mathbf{o}_G)\})}{\text{std}(\{R(\mathbf{q}, \mathbf{o}_1), \ldots, R(\mathbf{q}, \mathbf{o}_G)\})},$$

Figure 4: Illustration of the biases in GRPO. Note that the effective advantage of GRPO $a_{i,t}$ is equivalent to a reweighted version of the unbiased advantage $\tilde{A}_{i,t} = R(\mathbf{q}, \mathbf{o}_i) - \text{mean}(\mathbf{R})$. The terms $\text{std}(\mathbf{R})$ and $|\mathbf{o}_i|$ could bias the optimization by assigning different weights to different questions and responses, as denoted by the sizes of the blue circles and the lengths of the orange arrows. Upward arrows indicate positive advantages, and vice versa.

with the return $R(\mathbf{q}, \mathbf{o}_i)$ typically only including the *outcome verifiable reward* in LLM reasoning (the analysis also applies to process reward cases).

Compared to the objective function in Eq. (2), GRPO introduces two biases (see also Fig. 4):

- **Response-level length bias**: This arises from dividing by $|\mathbf{o}_i|$. For positive advantages ($\hat{A}_{i,t} > 0$, indicating a correct response), this bias results in greater gradient updates for shorter responses, leading the policy to favor brevity in correct answers. Conversely, for negative advantages ($\hat{A}_{i,t} < 0$, indicating an incorrect response), longer responses are penalized less due to their larger $|\mathbf{o}_i|$, causing the policy to prefer lengthier responses among incorrect ones.

- **Question-level difficulty bias**: This is caused by dividing the centered outcome reward by $\text{std}(\{R(\mathbf{q}, \mathbf{o}_1), \ldots, R(\mathbf{q}, \mathbf{o}_G)\})$. Questions with lower standard deviations (e.g., those that are too easy or too hard, with the outcome rewards being almost all 1 or 0) are given higher weights during policy updates. While advantage normalization is a common trick in RL (Andrychowicz et al., 2021), it is typically computed across an entire batch. In contrast, question-level normalization results in varying weights in the objective for different questions, leading to a difficulty bias in optimization.

**Length Bias Also Exists in Open-Source PPO Implementations**. We also examined several popular open-source implementations of vanilla PPO algorithms for LLM post-training. To our surprise, all of these implementations normalize the loss by response length (see listing 1 and Table 2), which **misaligns** with the PPO objective as defined in Eq. (2). This formulation-implementation misalignment was present even before the publication of GRPO. We speculate that the misalignment might originate from the *pretraining stage* (Shoeybi et al., 2019), where all tokens are packed into a fixed-length context and normalizing the loss by the context length (i.e., computing `loss.mean(-1)`) improves the numerical stability. However, in the *RL-tuning stage*, typical implementations (von Werra et al., 2020) normalize the loss by the response length, which is **not** a constant, introducing an unintended length bias.

Listing 1: Comparison between a typical open-source PPO loss implementation that is biased (red) and our implementation (green). `MAX_TOKENS` is a global constant during the entire training (unless budget curriculum is enabled), which specifies the maximum number of generation tokens. Other constants also work with differences in gradient norm.

```
def masked_mean(tensor, mask, dim):
-     return (tensor * mask).sum(axis=dim) / mask.sum(axis=dim)
+     return (tensor * mask).sum(axis=-1) / MAX_TOKENS

ppo_loss = ...        # compute per-token ppo loss
response_mask = ...   # per-token response mask
# per-response length normalization (e.g., OpenRLHF)
loss_variant1 = masked_mean(ppo_loss, response_mask, dim=-1).mean()
# OR per-batch length normalization (e.g., trl, verl)
loss_variant2 = masked_mean(ppo_loss, response_mask, dim=None).mean()
```

## 3.2 Dr. GRPO: Group Relative Policy Optimization Done Right

To avoid the aforementioned optimization bias in GRPO, we propose to simply remove the $\frac{1}{|\mathbf{o}_i|}$ and $\text{std}(\{R(\mathbf{q}, \mathbf{o}_1), \ldots, R(\mathbf{q}, \mathbf{o}_G)\})$ normalization terms. Meanwhile, to faithfully

| Repository | Code Link | Unbiased? |
|---|---|---|
| trl (von Werra et al., 2020) | PPO Loss | ✗ |
| OpenRLHF (Hu et al., 2024) | PPO Loss | ✗ |
| verl (Sheng et al., 2024) | PPO Loss | ✗ |
| SimpleRL-Zero (Zeng et al., 2025) | PPO Loss | ✗ |
| Open-Reasoner-Zero (Hu et al., 2025) | PPO Loss | ✗ |

Table 2: Many open-sourced PPO implementations contain length bias.

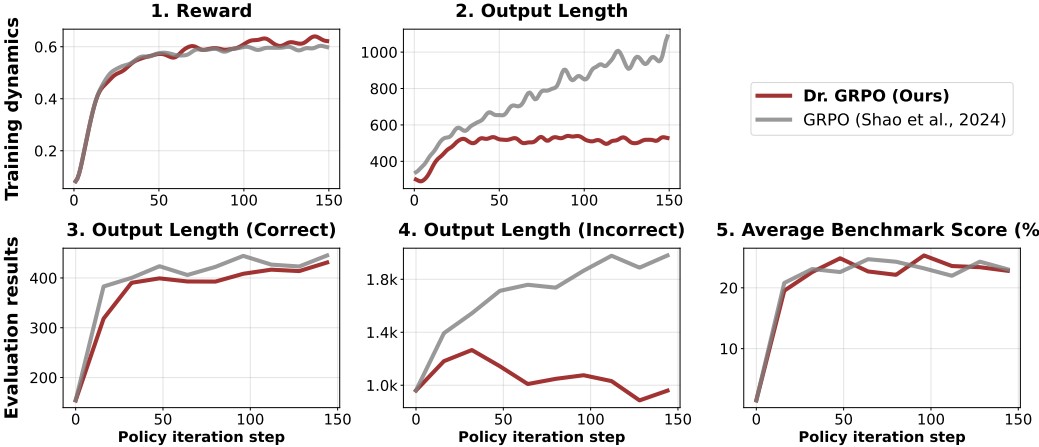

Figure 5: Comparison of Dr. GRPO and GRPO in terms of training dynamics (Top) and evaluation results (Bottom).

implement the unbiased optimization objective, we could replace the `mask.sum(axis=dim)` with a constant value (e.g., generation budget) in the `masked_mean` function in listing 1, as highlighted by the line in green. Notably, these simple modifications recover the PPO objective in Eq. (2), with the advantage estimated by Monte Carlo return with an unbiased baseline (Sutton & Barto, 2018). We give detailed derivations in App. A. We refer to our new optimization algorithm as **Dr. GRPO**. We next experimentally validate its effectiveness.

**Experimental settings**. We implement our algorithm using Oat (Liu et al., 2025a), a modular, research-friendly and efficient LLM RL framework. We adopt the Qwen2.5-1.5B base model and the R1 template (Template 1) for online RL-tuning. We implement the verification-based reward function using Math-Verify[2], with the following minimalistic rule:

$$R(\mathbf{q}, \mathbf{o}) = \begin{cases} 1 & \text{if } \mathbf{o} \text{ contains the correct final answer to } \mathbf{q} \\ 0 & \text{otherwise} \end{cases}$$

We run RL on questions sampled from the MATH (Hendrycks et al., 2021) training dataset, and compare the vanilla GRPO with the proposed Dr. GRPO. We evaluate the online model on five benchmarks: AIME2024, AMC, MATH500, Minerva Math and OlympiadBench. More experimental details including hyperparameters can be found in App. G.

**Results**. We report various metrics in Fig. 5 to demonstrate that Dr. GRPO can effectively mitigate the optimization bias and lead to **better token efficiency**. In particular, we first note that both GRPO and Dr. GRPO exhibit similar trend to DeepSeek-R1-Zero (Guo et al., 2025), namely their response length increases along with training reward (Plots 1 & 2). However, we observe that GRPO tends to continually generate longer responses even when the reward improvement slows down (Plot 2). Although such a phenomenon is often referred to as the "emergence" of long-CoT through RL (Zeng et al., 2025; Hu et al., 2025), we argue that

---

[2]https://github.com/huggingface/Math-Verify.

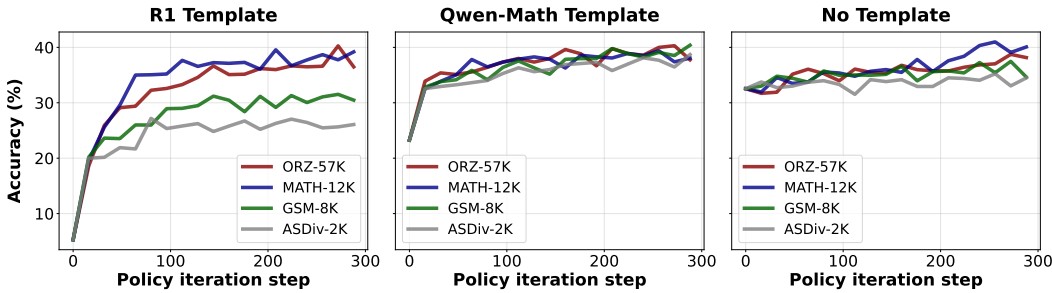

Figure 6: The average benchmark accuracy of different {template, question set} combinations during RL training.

it is also confounded by the response-level length bias (Sec. 3.1) during optimization[3]. In contrast, by computing the unbiased policy gradients, Dr. GRPO prevents the response length from growing wildly during training (Plot 2). Moreover, on evaluation benchmarks, the length of incorrect responses is substantially reduced by Dr. GRPO compared to the baseline (Plot 4), suggesting that an unbiased optimizer also **mitigates overthinking** (Chen et al., 2024).

### 3.3 A Duet of Template and Question Set Coverage in RL dynamics

Recall that the Qwen2.5-Math base models can readily answer questions with high accuracy without any prompt template (Sec. 2.2). Based on this intriguing observation, we are interested in how different templates affect the RL training. Furthermore, given the general belief that larger question set coverage leads to better performance (Luo et al., 2025; Hu et al., 2025), we also study the interaction between different templates and different levels of question coverage.

**Experimental settings**. Starting from the Qwen2.5-Math-1.5B base model, we apply R1 template, Qwen-Math template and No template respectively to run RL using Dr. GRPO. All experiments are repeated for different question sets that are detailed in Table 3.

| Question set | # | Description |
|:---:|:---:|:---:|
| ORZ | 57k | Combining AIME, Numina-Math, Tulu3 MATH; diverse and large amount |
| MATH | 12k | High-school math competition questions |
| GSM | 8k | Simpler grade-school math questions |
| ASDiv | 2k | Basic algebra $(+ - \times \div)$ questions |

Table 3: Different question sets that have different levels of difficulty and coverage.

**Results**. Fig. 6 shows the RL curves of different runs, from which we can make several interesting observations: **1)** Templates determine the performance of the initial policies, but RL can improve all policies to a comparable performance of $\sim 40\%$ (given a proper question set); **2)** When using the R1 template, question sets have a significant impact on the dynamics of RL, with too narrow coverage leading to lower plateau performance. However, when using the Qwen-Math template, the best final performance is attained by RL on GSM-8K, demonstrating that training on much simpler (and o.o.d.) questions can largely improve (nearly double) the test accuracy on harder questions. From these observations, we draw the following insights:

- The Qwen2.5-Math-1.5B base model already possesses strong math-solving capabilities (see the starting point in the right plot of Fig. 6). **Applying templates in fact destroys** the capability before RL reconstructs it. This implies that we should be more conservative in claiming the huge gains brought about by pure RL.

---

[3]We note that both Zeng et al. (2025) and Hu et al. (2025) employ PPO, which is unbiased by formulation. However, their loss implementations still introduce the length bias (see listing 1).

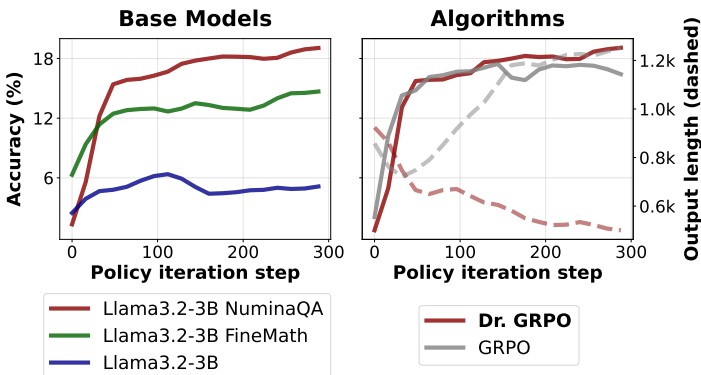

Figure 7: **Left**: The average benchmark performance curves of different base models. **Right**: The comparison between Dr. GRPO and GRPO with respect to reasoning accuracy (solid lines) and model response length (dashed lines).

- When there is a large **mismatch** between base models and templates (e.g., R1 template mismatches Qwen2.5-Math-1.5B), the policy improvement mainly comes from RL-tuning, thus requiring question set to have good coverage (left plot of Fig. 6). **Otherwise**, even a small and completely o.o.d. question set could induce the reasoning ability equally well, by **reinforcing useful reasoning behaviors instead of infusing new knowledge**.

### 3.4 Domain-Specific Pretraining Improves RL Ceiling

Recent successful R1-Zero-like replications of math reasoners mostly employ Qwen2.5 base models as the initial policies (Zeng et al., 2025; Cui et al., 2025; Hu et al., 2025), which are already strong math solvers and exhibit self-reflection patterns (Sec. 2.2 and 2.3). In this section we hope to explore the other side: **can R1-Zero-like training succeed on originally weak (in terms of math reasoning) base models?** We answer this question affirmatively, with the observation that *math pretraining would improve the ceiling of RL*.

**Experimental settings**. We adopt the Llama-3.2-3B base model as our starting point, and use the unbiased Dr. GRPO algorithm for RL-tuning with the R1 template. We hypothesize that domain-specific pretraining would help RL, hence we adopt the *Llama-3.2-3B-FineMath*[4], which is continual pretrained on the FineMath dataset (Allal et al., 2025). Moreover, as we hypothesize that Qwen2.5 models are likely to be pretrained on concatenated question-response texts (Sec. 2.2), we similarly prepare a concatenated dataset from NuminaMath-1.5 (Li et al., 2024b), and continual pretrain Llama-3.2-3B-FineMath for 2 epochs with learning rate 1e-5. We refer to the concatanated continual pretrained model as *Llama-3.2-3B-NuminaQA*.

**Results**. We present the RL curves of different base models in the left plot of Fig. 7. We observe that RL can even improve the vanilla Llama base model, but the gain is minimal. After continual pretraining (and concatenated continual pretraining) to embed math domain knowledge, Llama models can show much stronger RL performance, validating our hypothesis. We also revisit the GRPO's optimization bias with the Llama base model. The right plot of Fig. 7 compares the model performance and response length trained with GRPO and Dr. GRPO. We can clearly see that GRPO can produce the "double-increase" phenomenon, potentially leading to a **misperception** that long-CoT can also emerge on Llama models after math pretraining. Unfortunately, the increase of length might be due to the optimization bias (Sec. 3.1), which can be effectively mitigated by the proposed Dr. GRPO (Sec. 3.2 & right plot of Fig. 7).

## 4 Closing Remarks

We have taken a critical perspective to examine base models used for R1-Zero-like training, as well as algorithms used for RL. Through the analysis, we demystified how pretraining

---

[4]https://huggingface.co/HuggingFaceTB/FineMath-Llama-3B.

biases influence RL outcomes and how optimization choices, like GRPO, can unintentionally shape model behavior. With the proposed Dr. GRPO, we offer a simple fix that improves token efficiency while preserving reasoning performance. Our results show that scaling RL can be both effective and efficient—sometimes, less really is more.

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

# A  Policy Gradient Derivations

In the context of RL for LLM post-training, we typically maximize the value of

$$\mathcal{J}(\pi_\theta) = \mathop{\mathbb{E}}_{\mathbf{q} \sim p_\mathcal{Q}} \left[ \mathop{\mathbb{E}}_{\mathbf{o} \sim \pi_\theta(\cdot|\mathbf{q})} [R(\mathbf{q}, \mathbf{o})] \right], \tag{4}$$

where $R(\mathbf{q}, \mathbf{o}) = \sum_{t=1}^{|\mathbf{o}|} r(\mathbf{q}, \mathbf{o}_{\leq t})$ is the return (Sutton & Barto, 2018) of the trajectory $\mathbf{q}; \mathbf{o}$, and $r(\mathbf{q}, \mathbf{o}_{\leq t})$ represents the token-level reward for $t$-th token in response $\mathbf{o}$.

The Monte Carlo policy gradient (Sutton & Barto, 2018) of Eq. (4) is

$$\begin{aligned}
\nabla_\theta \mathcal{J}(\pi_\theta) &= \mathop{\mathbb{E}}_{\mathbf{q} \sim p_\mathcal{Q}} \left[ \mathop{\mathbb{E}}_{\mathbf{o} \sim \pi_\theta(\cdot|\mathbf{q})} [\nabla_\theta \log \pi_\theta(\mathbf{o}|\mathbf{q}) R(\mathbf{q}, \mathbf{o})] \right] \\
&= \mathop{\mathbb{E}}_{\mathbf{q} \sim p_\mathcal{Q}} \left[ \mathop{\mathbb{E}}_{\mathbf{o} \sim \pi_\theta(\cdot|\mathbf{q})} [\nabla_\theta \sum_{t=1}^{|\mathbf{o}|} \log \pi_\theta(o_t|\mathbf{q}, \mathbf{o}_{<t}) R(\mathbf{q}, \mathbf{o})] \right] \\
&= \mathop{\mathbb{E}}_{\mathbf{q} \sim p_\mathcal{Q}} \left[ \mathop{\mathbb{E}}_{\mathbf{o} \sim \pi_\theta(\cdot|\mathbf{q})} [\sum_{t=1}^{|\mathbf{o}|} \nabla_\theta \log \pi_\theta(o_t|\mathbf{q}, \mathbf{o}_{<t}) \sum_{t'=t}^{|\mathbf{o}|} r(\mathbf{q}, \mathbf{o}_{\leq t'})] \right] \\
&= \mathop{\mathbb{E}}_{\mathbf{q} \sim p_\mathcal{Q}} \left[ \mathop{\mathbb{E}}_{\mathbf{o} \sim \pi_\theta(\cdot|\mathbf{q})} \left[ \sum_{t=1}^{|\mathbf{o}|} \nabla_\theta \log \pi_\theta(o_t|\mathbf{q}, \mathbf{o}_{<t}) \left( \sum_{t'=t}^{|\mathbf{o}|} r(\mathbf{q}, \mathbf{o}_{\leq t'}) - B(\mathbf{q}, \mathbf{o}_{<t}) \right) \right] \right],
\end{aligned} \tag{5}$$

where $B(\mathbf{q}, \mathbf{o}_{<t})$ is a variance reduction term, which is invariant with respect to $o_t$ so that

$$\begin{aligned}
\mathop{\mathbb{E}}_{o_t \sim \pi_\theta(\cdot|\mathbf{q}, \mathbf{o}_{<t})} [\nabla_\theta \log \pi_\theta(o_t|\mathbf{q}, \mathbf{o}_{<t}) B(\mathbf{q}, \mathbf{o}_{<t})] &= \mathop{\mathbb{E}}_{o_t \sim \pi_\theta(\cdot|\mathbf{q}, \mathbf{o}_{<t})} [\nabla_\theta \log \pi_\theta(o_t|\mathbf{q}, \mathbf{o}_{<t})] B(\mathbf{q}, \mathbf{o}_{<t}) \\
&= [\sum_{o_t} \pi_\theta(o_t|\mathbf{q}, \mathbf{o}_{<t}) \nabla_\theta \log \pi_\theta(o_t|\mathbf{q}, \mathbf{o}_{<t})] B(\mathbf{q}, \mathbf{o}_{<t}) \\
&= [\sum_{o_t} \nabla_\theta \pi_\theta(o_t|\mathbf{q}, \mathbf{o}_{<t})] B(\mathbf{q}, \mathbf{o}_{<t}) \\
&= [\nabla_\theta \sum_{o_t} \pi_\theta(o_t|\mathbf{q}, \mathbf{o}_{<t})] B(\mathbf{q}, \mathbf{o}_{<t}) \\
&= [\nabla_\theta 1] B(\mathbf{q}, \mathbf{o}_{z<t}) = 0.
\end{aligned}$$

Typically, we set $B(\mathbf{q}, \mathbf{o}_{<t}) = \mathop{\mathbb{E}}_{\mathbf{o}_{\geq t} \sim \pi_\theta(\cdot|\mathbf{q}, \mathbf{o}_{<t})} [\sum_{t'=t}^{|\mathbf{o}|} r(\mathbf{q}, \mathbf{o}_{\leq t'})]$, which is the expected cumulative reward in the future (also known as the value of the current state), and denote $A(o_t|\mathbf{q}, \mathbf{o}_{<t}) = \sum_{t'=t}^{|\mathbf{o}|} r(\mathbf{q}, \mathbf{o}_{\leq t'}) - B(\mathbf{q}, \mathbf{o}_{<t})$ as the advantage. In the case of outcome reward, $\sum_{t'=t}^{|\mathbf{o}|} r(\mathbf{q}, \mathbf{o}_{\leq t'}) = \sum_{t=1}^{|\mathbf{o}|} r(\mathbf{q}, \mathbf{o}_{\leq t}) = R(\mathbf{q}, \mathbf{o})$.

By setting $B(\mathbf{q}, \mathbf{o}_{<t}) = \text{mean}(\{R(\mathbf{q}, \mathbf{o}_1), \dots, R(\mathbf{q}, \mathbf{o}_G)\})$, the policy gradient of Eq. (5) becomes

$$\nabla_\theta \mathcal{J}(\pi_\theta) = \mathop{\mathbb{E}}_{\mathbf{q} \sim p_\mathcal{Q}} \left[ \mathop{\mathbb{E}}_{\{\mathbf{o}_i\}_{i=1}^G \sim \pi_\theta(\cdot|\mathbf{q})} [\frac{1}{G} \sum_{i=1}^G \sum_{t=1}^{|\mathbf{o}|} \nabla_\theta \log \pi_\theta(o_{i,t}|\mathbf{q}, \mathbf{o}_{i,<t}) \tilde{A}_{i,t}] \right], \tag{6}$$

where

$$\tilde{A}_{i,t} = \frac{R(\mathbf{q}, \mathbf{o}_i) - \text{mean}(\{R(\mathbf{q}, \mathbf{o}_1), \dots, R(\mathbf{q}, \mathbf{o}_G)\})}{\color{red}{\cancel{\text{std}(\{R(\mathbf{q}, \mathbf{o}_1), \dots, R(\mathbf{q}, \mathbf{o}_G)\})}}}.$$

We adopt the PPO (Schulman et al., 2017b) objective to compute Eq. (6):

$$\mathcal{J}(\pi_\theta) = \mathbb{E}[\mathbf{q} \sim p_\mathcal{Q}, \{\mathbf{o}_i\}_{i=1}^G \sim \pi_{\theta_{old}}(\cdot|\mathbf{q})]$$

$$\frac{1}{G} \sum_{i=1}^G \color{red}{\cancel{\frac{1}{|\mathbf{o}_i|}}} \color{black}{\sum_{t=1}^{|\mathbf{o}_i|} \left\{ \min \left[ \frac{\pi_\theta(o_{i,t}|\mathbf{q}, \mathbf{o}_{i,<t})}{\pi_{\theta_{old}}(o_{i,t}|\mathbf{q}, \mathbf{o}_{i,<t})} \tilde{A}_i, \text{clip} \left( \frac{\pi_\theta(o_{i,t}|\mathbf{q}, \mathbf{o}_{i,<t})}{\pi_{\theta_{old}}(o_{i,t}|\mathbf{q}, \mathbf{o}_{i,<t})}, 1 - \epsilon, 1 + \epsilon \right) \tilde{A}_i \right] \right\}},$$

from which we conclude that both std and $|\mathbf{o}|$ should not appear in the RL objective.

**Unbiasedness of $\tilde{A}_{i,t}$.** We note that $\tilde{A}_{i,t}$ computed above is equivalent to that of REINFORCE Leave-One-Out (RLOO) (Ahmadian et al., 2024; Kool et al., 2019) up to a scaling factor, which can be subsumed into the learning rate without affecting the RL dynamics. Specifically,

$$\frac{G}{G-1} \cdot \tilde{A}_{i,t} = \frac{G}{G-1} R(\mathbf{q}, \mathbf{o}_i) - \frac{G}{G-1} \frac{1}{G} \sum_{j=1}^{G} R(\mathbf{q}, \mathbf{o}_j)$$

$$= \frac{G}{G-1} R(\mathbf{q}, \mathbf{o}_i) - \frac{1}{G-1} \sum_{j=1, j \neq i}^{G} R(\mathbf{q}, \mathbf{o}_j) - \frac{1}{G-1} R(\mathbf{q}, \mathbf{o}_i)$$

$$= \hat{A}_{i,t}^{\text{RLOO}}.$$

## B  Detailed Benchmark Results

We show the detailed benchmark results for three scales (1.5B, 3B and 7B) in Table 4. We also include the instruct models at the same scale and R1-Distill models for comparison. Note that since we employ the Qwen2.5-Math base models, which have a context length of 4k, we thus limit the generation budget at 3k for all baselines compared. For models that are trained for a longer context (OpenReasoner-Zero end R1-Distill-Qwen), we also report their performance at 8k generation budget.

| Base model + Method | AIME24 | AMC | MATH500 | Minerva | OlympiadBench | Avg. |
|---|---|---|---|---|---|---|
| Qwen2.5-Math-1.5B | **20.0** | 32.5 | 33.0 | 12.5 | 22.8 | 24.2 |
| Qwen2.5-Math-1.5B* | 16.7 | 43.4 | 61.8 | 15.1 | 28.4 | 33.1 |
| *Oat-Zero-1.5B* | **20.0** | **53.0** | **74.2** | **25.7** | **37.6** | **42.1** |
| R1-Distill-Qwen-1.5B @ 3k | 2.5 | 21.7 | 52.2 | 16.3 | 17.3 | 22.0 |
| R1-Distill-Qwen-1.5B @ 8k | 20.0 | 49.4 | 77.4 | 25.0 | 35.8 | 41.5 |
| Qwen2.5-Math-1.5B-Instruct | 10.0 | 48.2 | 74.2 | 26.5 | 40.2 | 39.8 |
| Llama-3.2-3B | 0.0 | 2.4 | 6.4 | 6.3 | 1.3 | 3.3 |
| + RL w. Dr. GRPO | 3.3 | 7.2 | 10.0 | 11.0 | 2.2 | 6.8 |
| Llama-3.2-3B-FineMath | 0.0 | 3.6 | 18.4 | 5.9 | 2.2 | 6.0 |
| + RL w. Dr. GRPO | 3.3 | 10.8 | 38.0 | 12.9 | 9.0 | 14.8 |
| Llama-3.2-3B-NuminaQA | 0.0 | 0.0 | 0.6 | 0.0 | 0.1 | 0.14 |
| + RL w. Dr. GRPO (*Oat-Zero-3B*) | **6.7** | **18.1** | **50.0** | **14.3** | **14.7** | **20.7** |
| Llama-3.2-3B-Instruct | 6.7 | 15.7 | 38.8 | 11.8 | 12.6 | 17.1 |
| Qwen2.5-Math-7B | **16.7** | 38.6 | 50.6 | 9.9 | 16.6 | 26.5 |
| Qwen2.5-Math-7B* | 0.2 | 45.8 | 69.0 | 21.3 | 34.7 | 38.2 |
| SimpleRL-Zero-7B | 26.7 | 60.2 | 78.2 | 27.6 | 40.3 | 46.6 |
| PRIME-Zero-7B | 16.7 | **62.7** | **83.8** | **36.0** | 40.9 | 48.0 |
| OpenReasoner-Zero-7B @ 3k | 13.3 | 47.0 | 79.2 | 31.6 | 44.0 | 43.0 |
| OpenReasoner-Zero-7B @ 8k | 13.3 | 54.2 | 82.4 | 31.6 | **47.9** | 45.9 |
| *Oat-Zero-7B* | **43.3** | **62.7** | 80.0 | 30.1 | 41.0 | **51.4** |
| R1-Distill-Qwen-7B @ 3k | 10.0 | 26.2 | 60.1 | 23.0 | 23.1 | 28.5 |
| R1-Distill-Qwen-7B @ 8k | 33.3 | 68.4 | 88.1 | 35.9 | 47.7 | 54.7 |
| Qwen2.5-Math-7B-Instruct | 16.7 | 53.0 | 83.6 | 29.8 | 42.7 | 45.1 |

Table 4: A comparison on benchmark scores. *Ours* models are RL-tuned by our minimalist recipe (Sec. 1). * means we employ the best template (no template) to generate answers, such that the test scores are highest and can faithfully reflect the capabilities of the base models.

## C  Extended Empirical Results

In this section we present two extended empirical results for (1) the ablation of different bias terms in GRPO and (2) statistical significance of Dr. GRPO's results. We RL-tune the

Qwen2.5-1.5B base model on a mixture of 3K diverse math questions drawn from ASDiv, MATH, and AIME (pre-2023).

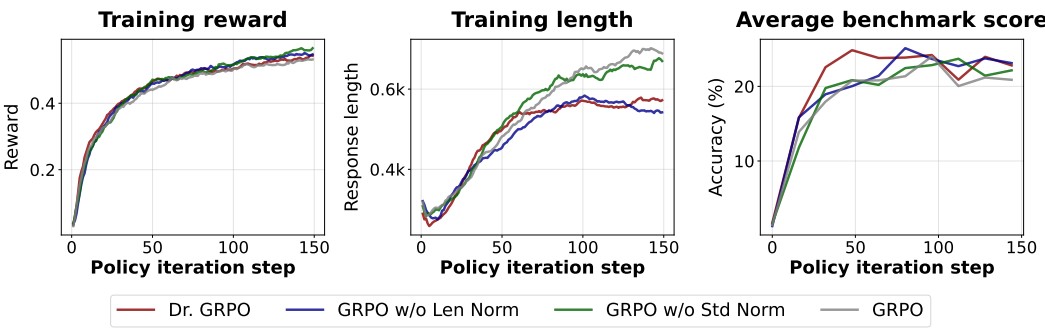

Figure 8: Ablation results on the two bias terms in GRPO.

Fig. 8 shows the training and evaluation curves for the following variants: Dr. GRPO, GRPO w/o length normalization, GRPO w/o standard deviation (std) normalization and Vanilla GRPO. From the middle subplot, we observe that both Dr. GRPO and the variant without length normalization generate shorter responses compared to the other two. This confirms that the length bias term has a more significant influence on response length–consistent with our expectations.

In terms of performance, Dr. GRPO and the other ablated variants consistently outperform vanilla GRPO in both training rewards and evaluation accuracy. This indicates that removing bias terms (either length or std) improves policy learning, validating our motivation for Dr. GRPO.

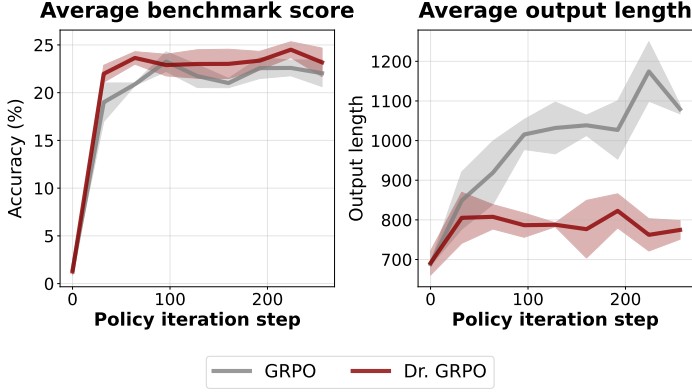

Figure 9: Evaluation results of 3 independent RL runs. The mean curves are drawn in solid lines and the standard deviation is plotted in the shaded areas.

Fig. 9 compares GRPO and Dr. GRPO across three independent runs. We observe that Dr. GRPO consistently demonstrates statistically significant improvements–both in token efficiency and final accuracy–across different random seeds.

## D   Keyword-based Detection and LLM-Based Identification of Self-Reflection Behaviors

We construct a pool of carefully selected keywords and phrases that signal self-reflection behaviors in the LLM's responses. However, LLM-generated responses often contain hallucinations and off-topic content, leading to the presence of simple, ambiguous keywords that do not necessarily indicate genuine self-reflection. For instance, terms like "wait"

and "try again" frequently result in false positive detections. To reduce false positives, we maintain a small, highly selective keyword pool consisting of terms that are strongly indicative of self-reflection. In our experiment, the keyword pool is limited to: recheck, rethink, reassess, reevaluate, re-evaluate, reevaluation, re-examine, reexamine, reconsider, reanalyze, double-check, check again, think again, verify again, and go over the steps.

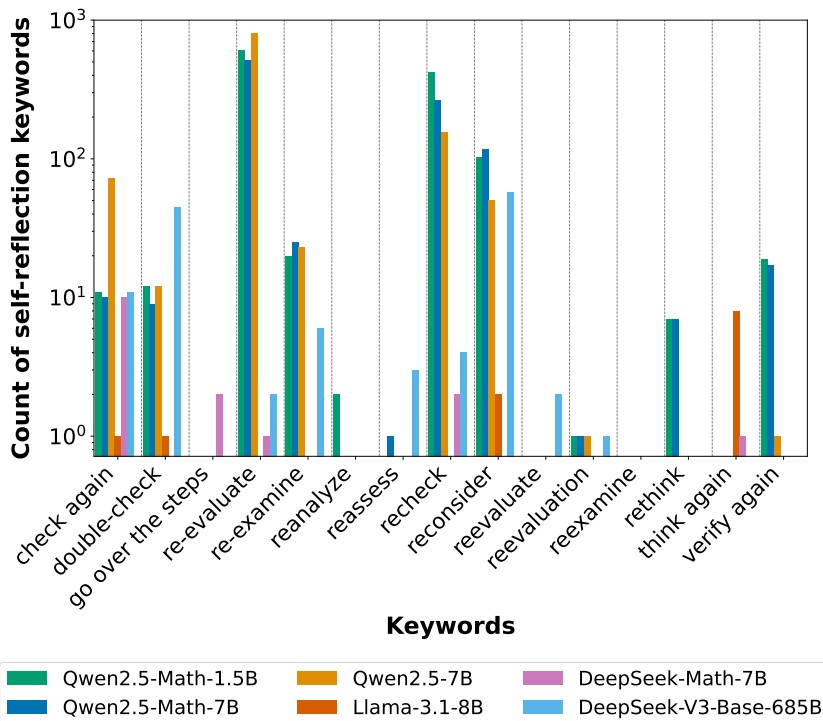

Figure 10: Count of keyword occurrences out of 40,000 responses (500 questions × 8 responses per question × 10 temperatures). y is in log scale.

We present the occurrences of various keywords in the responses generated by different models in Figure 10. Interestingly, different model families emphasize different keywords. For instance, phrases such as "check again", "double-check", "re-evaluate", "re-examine", "recheck", "reconsider", and "verify again" appear most frequently in the Qwen2.5 family. In contrast, "re-evaluate", "re-examine", and "verify again" do not appear in the responses of the DeepSeek family, while Llama models frequently use the phrase "think again." We hypothesize that this phenomenon results from differences in the pretraining data, particularly in relation to reasoning and mathematics.

Although we meticulously select the keyword pool, it may still be insufficient to identify some implicit behaviors of self-reflection that do not contain a specific keyword. Additionally, it can lead to false positives, as illustrated in Case (a) of Figure 11. To address these limitations and more accurately assess the self-reflection capability of base models, we leverage stronger LLMs (GPT-4o-mini in our experiments) to analyze the responses and determine whether they exhibit explicit self-reflection (e.g., keywords like "recheck" and "reevaluate") or implicit self-reflection (e.g., more sophisticated patterns that cannot be easily captured through keyword matching). This approach helps distinguish true self-reflection behaviors from superficial or incidental use of related terms.

While LLM-based detection effectively filters out false positives from keyword-based detection and identifies implicit self-reflection behaviors, it can still misclassify responses, particularly when they are lengthy and complex. For instance, Case (b) in Figure 11 shows a false positive in LLM-based detection, where the response is categorized as self-reflection by the LLM but does not actually exhibit self-reflection. This type of error can be filtered out by keyword-based detection. To enhance robustness, we integrate keyword-based

| **Case(a):** What is the smallest positive multiple of 23 that is 4 more than a multiple of 89? | **Case(b):** What is the greatest multiple of 3 that is less than 500 and is also a multiple of 4? |
|---|---|
| **Response**

To find the smallest positive multiple of 23 that, ...

Start with the smallest possible value of $k$ by checking if they satisfy the equation. If not, increase $k$ and check again.

...

Now, trying $k = 21$: $23 \cdot 21 = 483$ ...

... | **Response**

To find the greatest multiple of 3 that is less than 500 and is also a multiple of 4, I need to first consider the greatest multiple of 4 less than 500, which is 496. However, 496 is not a multiple of 3. I then need to consider the next multiple of 4, which is 492. 492 is a multiple of 3, since 492 / 3 = 164 with no remainder.</think>
<answer> 492 <answer> |

Figure 11: **Case (a)**: a false positive in keyword-based detection. **Case (b)**: a false positive in LLM-based detection.

and LLM-based detection through cross-validation. The combined detection results, along with the individual results from keyword-based and LLM-based methods, are presented in Figure 12.

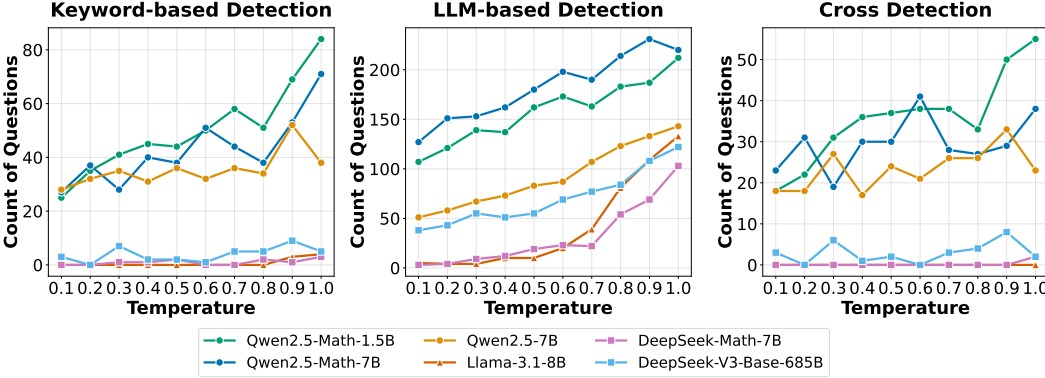

Figure 12: Comparison of keyword-based detection, LLM-based detection, and cross detection. Self-reflections are counted at the question level across 500 questions, where a question is marked as having self-reflection if at least one of its eight responses exhibits self-reflection.

# E  Examples of Aha Moment in DeepSeek-V3-Base

Fig. 13 shows two examples to demonstrate that the DeepSeek-V3-Base model already exhibits the so-called "aha moment" even before the RL-tuning.

# F  Comparison Between DeepSeek-V3-Base and DeepSeek-R1-Zero

We analyze DeepSeek-V3-Base and DeepSeek-R1-Zero to understand changes in model behavior during R1-Zero training. In Fig. 14, we present the breakdown of response categories across difficulty levels for 500 MATH questions evaluated on both models. The results indicate that most incorrect responses are corrected after RL training, demonstrating substantial performance gains from R1-Zero training. Meanwhile, we find an increase in unformatted responses, which aligns with the observation in Liu et al. (2025b).

In Table 5, we report the average response lengths across categories. Note that truncated responses would fall into any of the other three categories if a larger context size were used; thus, we exclude them from the table. The results show a substantial increase in response lengths across all categories, including correct responses, consistent with the results in the

| **Question(a):** An element is randomly chosen from among the first 15 rows ... | **Question(b):** Find the least positive integer $n$ such that $1/\sin 45° \sin 46° + ...$ |
|---|---|
| **Response** | **Response** |
| In Pascal's Triangle, every row starts and ends with 1, ... | First, I notice that the left-hand side of the equation has the form ... |
| ... | Simplifying, I get ... |
| This can be calculated as: *awkward silence* Wait, I'm overthinking. Let's try again. The number of elements in the first n rows of Pascal's Triangle is $(n+1)!/[2! * (n-1)!]$. | Aha! I can use this to get $\sin(a+2b) = \sin((a+b)+b) = 3\sin(a+b) ...$ This looks messy, but I notice that it has the form $\sin a \cos b + \cos a \sin b$ again. This gives me an idea. Maybe I can ... |
| ... | ... |

Figure 13: Cases showing that DeepSeek-V3-Base already exhibits "Aha moment" even before RL tunning.

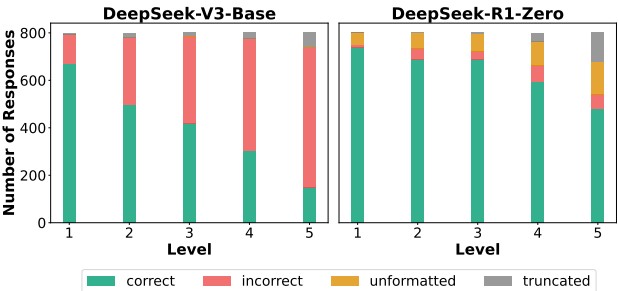

| Category | Base | R1-Zero |
|---|---|---|
| Unformatted | 880.7 | 7870.3 |
| Correct | 621.3 | 4965.4 |
| Incorrect | 1038.9 | 8206.1 |

Table 5: Average response string lengths across categories for DeepSeek-V3-Base (Base) and DeepSeek-R1-Zero (R1-Zero).

Figure 14: Breakdown of response categories across difficulty levels in the MATH dataset for DeepSeek-V3-Base and DeepSeek-R1-Zero.

Fig. 3 of Guo et al. (2025). However, the average length of incorrect responses is notably longer than that of correct responses. We hypothesize this is because more challenging questions generally require longer responses due to increased reasoning complexity, and incorrect responses are more likely to originate from harder questions, resulting in a longer average length.

**Self-reflection does not necessarily imply higher accuracy.** To investigate whether self-reflection behaviors are associated with model performance during the inference (acknowledging that self-reflection may improve exploration during training—a potential positive effect outside this section's scope), we analyze questions that elicit at least one response with self-reflection from DeepSeek-R1-Zero across eight trials. For each question, we sample 100 responses and divide them into two groups: those with self-reflection and those without. We then compute the accuracy difference between these two groups for each question. As shown in Fig. 15, the results indicate that nearly half responses with self-reflection do not achieve higher accuracy than those without self-reflection, suggesting that self-reflection does not necessarily imply higher inference-stage accuracy for DeepSeek-R1-Zero.

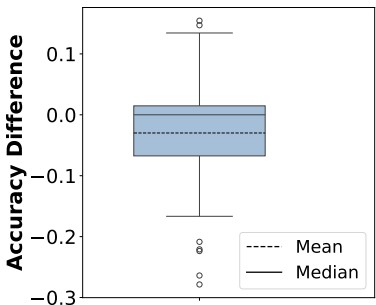

Figure 15: Accuracy difference between responses with and without self-reflection for each question (responses sampled from DeepSeek-R1-Zero).

# G  Detailed Experimental Settings

All our experiments are performed on $8 \times$ A100 GPUs and finished in about one day. We enable the actor-learner collocation supported by Oat (Liu et al., 2025a) to optimize the training efficiency. We show the experimental configurations in Table 6.

| Parameter | Value |
|---|---|
| \multicolumn ACTOR | |
| Maximum response length | 3000 tokens |
| Sampling temperature | 1.0 |
| (top P, top k) | (1.0, -1) |
| Number of responses per question | 8 |
| \multicolumn LEARNER | |
| Optimizer | AdamW |
| Adam parameters $(\beta_1, \beta_2)$ | (0.9, 0.95) |
| Weight decay | 0.0 |
| Gradient norm clipping | 1.0 |
| Learning rate scheduler | Constant |
| Learning rate | $1 \times 10^{-6}$ |
| Inner proximal update epoch | 1 |
| KL loss coefficient | 0.0 |
| KL penalty coefficient | 0.0 |
| Policy clipping parameter | 0.2 |

Table 6: Hyperparameter configurations used in all experiments.

# H  Prompts Used for GPT-As-A-Judge

Prompt for checking the model's question-answering ability.

> **Prompt for Checking Question-Answering Ability**
>
> I will send you a question and a long response generated by an LLM. Your task is to determine whether the output attempts to answer the question or not. The output may sometimes include irrelevant content, hallucinations, or random, off-topic responses.
> Please classify the output into one of the following categories:
> **Output Format**:
> Your response must start with a **single integer** (0 or 1), followed by a **brief explanation**.
>
> - **Return 0:** → The output is not trying to answer the question (e.g., irrelevant content, random talking, hallucinations). *Example output:* '0: The response is off-topic and does not address the question.'
>
> - **Return 1:** → The output attempts to answer the question, regardless of how complete or accurate the answer is. *Example output:* '1: The response engages with the question, even if the answer is incomplete or incorrect.'
>
> **Question:** {question}
> **Response:** {response}

Prompt for LLM-based detection to determine whether a response contains self-reflection behaviors.

---

**LLM-based Detection for Self-Reflection**

I will send you a mathematical question along with a detailed response. Your task is to determine whether the response is attempting to answer the question. If the response is off-topic, hallucinated, random talk, or otherwise irrelevant, mark it as **0**. Otherwise, assess whether the response exhibits self-reflection.
**Categorization Rules**:

1. **Category 0**: The response is **off-topic, nonsensical, incoherent, overly repetitive, or lacks logical reasoning**.
   - Example cases:
     - The response does not relate to the question.
     - It contains meaningless or hallucinated content.
     - It consists of excessive repetition without coherence.

2. **Category 1**: The response **attempts to answer the question** but does **not** exhibit self-reflection.
   - Example cases:
     - The response directly solves the problem without revisiting steps.
     - No attempt is made to verify the correctness of the answer or explore alternative solutions.

3. **Category 2**: The response **demonstrates self-reflection** at any level.
   - This may include:
     - **Explicit self-reflection keywords**, such as: *recheck, rethink, reassess, reevaluate, re-evaluate, reevaluation, re-examine, reexamine, reconsider, reanalyze, double-check, check again, think again, verify again, go over the steps*, etc.
     - **Implicit self-reflection behaviors**, such as revisiting the solution, questioning assumptions, or considering alternative approaches **without explicit keywords**.
   - If any form of self-reflection is present, **always categorize it as 2**, regardless of correctness or answer quality.

4. **Category 3**: The response consists **solely of Python code for calculations** without exhibiting self-reflection.
   - Example cases:
     - The response only provides a Python script to compute the solution **without any verification, re-evaluation, or alternative considerations**.

**Output Format**:
Your response should first provide a **very brief explanation** of your analysis, followed by a **single category number (0, 1, 2, or 3)** at the end. You must include the category number at the end of your response.
**Example outputs:**

- 'The response is off-topic and does not attempt to answer the question. 0.'
- 'The response provides a direct solution without self-reflection. 1.'
- 'The response demonstrates self-reflection. 2.'
- 'The response consists solely of Python code without any self-reflection. 3.'

**Question:** {question}
**Response:** {response}

---

