# OpenReview forum: "Understanding R1-Zero-Like Training: A Critical Perspective"
_colmweb.org/COLM/2025/Conference — COLM 2025_

### Official Review · Reviewer_dTZp · 2025-05-13

**Rating:** 9
**Confidence:** 4
**Ethics Flag:** 1

**Summary:**

This paper takes apart the R1-Zero recipe and studies the influence of base model and RL algorithm onto the final reasoning capabilities. It finds that base models already have reasoning capabilities and that the GRPO algorithm contains length bias that encouraged longer outputs and thereby more reasoning. The paper implements a correction and finds that it effectively improves RL optimization. Finally, the authors combine their insights about base model capabilities and algorithmic bias to establish a minimal R1-Zero recipe.

**Reasons To Accept:**

- The paper is fun to read because it is inquisitive and informative. There are lots of bits to learn about existing models, the way they are compared and tuned, and about RL algorithms.
- The writing is clear and concise, and findings well visualized.
- The experiments clearly and thoroughly isolate the questions under study.
- It shows that small oversights here and there can have a butterfly effect leading to conclusions of new capabilities. It is worth reflecting over how such oversight can happen, and how it can be prevented in future studies. It invites a more critical perspective and demands more thorough (slower?) research, which imho the field is in dire need of.

**Reasons To Reject:**

-

---

> ### Author Response · Authors · 2025-06-03
>
> We sincerely appreciate your insightful comments and will continue refining the paper to further improve its quality. Thank you again for your time and thoughtful feedback.

---

### Official Review · Reviewer_pF1u · 2025-05-13

**Rating:** 10
**Confidence:** 4
**Ethics Flag:** 1

**Summary:**

This paper highlights the two fundamental components of training R1-Zero-like models: the base model and the reinforcement learning technique. It provides extensive insights, including how templates affect model performance, the importance of the base model's self-reflection ability, and the potential bias introduced by GRPO. The authors also propose a simple fix to GRPO, which produces unbiased responses while maintaining comparable performance.

**Questions To Authors:**

Given the argument regarding the length bias of GRPO, is it possible to design a normalization term that depends not only on the length but also on the reward value?

**Reasons To Accept:**

- In my opinion, this paper provides valuable insights into training R1-Zero-like models. All the findings are supported by rigorous experimental results with clear explanations. I enjoyed reading the paper.
- The finding that “Aha Moment Already Appears in Base Models Including DeepSeek-V3-Base” is particularly inspiring and insightful. It identifies the cause of the "aha moment" and suggests a potential direction for improving the model’s reasoning ability in the future: focusing on enhancing the base model rather than relying on additional reinforcement learning techniques.
- The finding of bias in GRPO is reasonable. It is somewhat surprising that simply removing certain terms does not sacrifice performance and can even reduce overthinking.

**Reasons To Reject:**

I do not see any significant weaknesses in this paper. One minor issue is that Figure 5 overlaps with the caption of Table 2.

---

> ### Author Response · Authors · 2025-06-03
>
> We sincerely appreciate your encouraging comments and thoughtful questions. In our revision, we will adjust the layout to ensure proper alignment of Figure 5 and Table 2.
>
> Regarding your suggestion, we agree that it is indeed possible to design a normalization term that incorporates reward values. For instance, instead of dividing by the standard deviation as done in GRPO, one could *multiply* the advantage by the standard deviation. This acts as a form of reward shaping, effectively upweighting the gradients of samples with higher uncertainty—i.e., those with approximately equal numbers of correct and incorrect responses—since they are more likely to benefit from improvement.
>
> While this idea lies beyond the scope of our current work, we find it promising and plan to explore it in future research.
>
> Thank you once again for your valuable feedback.

---

> > ### Comment · Reviewer_pF1u · 2025-06-06
> >
> > Thanks for the explanation!

---

### Official Review · Reviewer_FjK5 · 2025-05-17

**Rating:** 5
**Confidence:** 3
**Ethics Flag:** 1

**Summary:**

This work investigates the recent phenomenon of R1-style training to deconstruct it in order to improve it. First, they investigate base models and find that Qwen base already includes instruction-following abilities and Deepseek V3-base already includes self-reflection abilities. Next they argue the GRPO objective incorrectly averages by the response length and divides the advantage by the std of a group of responses. The experiments section train models, mainly Qwen, on math tasks. Based on experiments, it is argued that templates may be unnecessary for models with some amount of instruction-tuning and they may even be harmful to performance. Finally, they improve llama 3.2 3B with math pretraining to show that domain specific question-answer pretraining is key to leveraging RL.

**Questions To Authors:**

wouldn't Deepseek base also have to have seen something like instruction-following data in order to follow a template such as User: ... Assistant:.... Can you give an idea how you think Qwen different? Just more explicitly structured data?


in Section 3.2, you claim
> even a small and completely o.o.d. question set could induce the reasoning ability equally well, by reinforcing useful reasoning behaviors instead of infusing new knowledge.

do you have proof or a citation for that? it seems unrelated to your experiments on templates



What is the takeaway from the experiments with Llama 3.2? Is it that NuminaMath is the right choice for pretraining because of the question-answer format and the initial bad performance could be due to a bad template?

**Reasons To Accept:**

This paper presents a series of results that help illuminate modern r1-style training. This is especially useful as technical reports from companies often omit smaller details and open-source reproductions. Dr. GRPO is a good introspection of the GRPO method and has some potentially interesting theoretical insights. There are definitely interesting insights about R1 here, although they are quite scattered without a clear story.

**Reasons To Reject:**

The work is interesting but disorganized. It presents lots of separate results without a particular story. Still, I believe it is worth accepting if the method part of the work Dr. GRPO can be made a little stronger. As it stands, I believe the work has theoretical issues and it is not strongly

**response length bias may be wrong**

I believe the point on Response Length Bias is wrong but I am open to changing my mind
- GRPO (and PPO) are extensions of regular policy gradient
- according to the policy gradient theorem, the gradient $\nabla J(\theta)$ is not equal but *proportional* to $ \propto \sum_s \mu(s) \sum_a q_\pi (s,a) \grad \pi(a|s,\theta) $
- this proportionality is the average length of the episode (equation 13.5, Sutton and Barto, 2020)
- in the case of the MDP of token generation, we know the exact length of the episode, which is the number of tokens generated
- I believe this is the reason for averaging by the number of generated tokens

We don't see the corrected behaviour in Dr. GRPO
- removing averaging should increase the response length of correct answers
- but Figure 5 (bottom left) shows that the original GRPO has the longer correct response lengths
- the main practical reason for the averaging is that, without it, a model is incentivized to generate more tokens for a correct answer

**better justifying Dr. GRPO empirically**

Overall, given that there are two innovations here, it would be reasonable to do an ablation and show the effects of each one. A 1.5B run would be sufficient with just removing response length average and just removing dividing by std deviation.

Second, it is unclear whether Dr. GRPO actually improves results empirically or just reduces output length. The only claim is that it improves "token efficiency", which is to say it reduces the length of generations? What is the specific speedup for your experiments and how robust are they? i.e. do you have a mean + std over seeds?

---

> ### Author Response · Authors · 2025-06-03
>
> Thank you very much for your insightful comments and questions, which we address below:
>
> **Q1: Concerns about response length bias**
>
> We appreciate your observation. As stated in Sutton and Barto, 2020 (Equation 13.5 and subsequent text), “the constant of proportionality is the average length of an episode.”
>
> To clarify, this proportionality involves a **constant**, which is the **average episode length**. In the context of language MDPs, this corresponds to the ***average** response length*. However, original GRPO computes length normalization on a **per-response basis**, introducing a dynamic factor that varies with each sample and leads to **length bias**—e.g., policy gradients for longer responses are systematically downweighted.
>
> Dr. GRPO eliminates this bias by normalizing using a **constant**, which may be any fixed value. While we use `MAX_TOKEN_LENGTH` for convenience, using the **average length** would yield an equivalent result up to a scaling factor—consistent with the formulation in Sutton and Barto, 2020. Therefore, Dr. GRPO does **not** conflict with Equation 13.5.
>
> Furthermore, Appendix A provides an exact policy gradient derivation, offering a more precise analysis of the bias introduced in GRPO.
>
> **Q2: Why the "corrected" behavior is not visible in Dr. GRPO**
>
> Theoretically, GRPO’s bias favors shorter correct responses. However, this does not necessarily manifest in practice because correct responses must meet a **minimum length requirement** to fully solve the task (e.g., perform multi-step reasoning), which may not be reducible further. We hypothesize this is the reason for GRPO not being shorter than Dr.GRPO in subplot 3 of Figure 5.
>
> Conversely, GRPO's bias towards **longer incorrect responses** is more readily observable, as incorrect responses face fewer constraints and can easily become verbose or degenerate.
>
> This asymmetry is clearly illustrated in Figure 5 (subplots 3 & 4): while the average length of **correct** responses is similar across methods, GRPO produces significantly longer **incorrect** responses, consistent with its length bias.
>
>
> **Q3: Ablation and statistical significance**
>
> Thank you for the suggestion. We have initiated two additional runs:
> 1. An ablation to isolate the effects of length and standard deviation biases;
> 2. Multiple runs with different random seeds to assess statistical significance.
>
> We will update the results once training completes.
>
> **Q4: Unclear whether Dr. GRPO actually improves results empirically or just reduces output length**
>
> We would like to clarify that Dr. GRPO is designed to correct the implicit biases present in GRPO, effectively making it “done right.” While these biases are most evident in response length, they may not be as immediately visible in terms of absolute performance, since both GRPO and Dr. GRPO optimize for the same RL objective—answer correctness.
>
> That said, we did observe a slight improvement in reward with Dr. GRPO, as shown in Figure 6 (1), indicating that the correction not only removes bias but can also lead to marginal performance gains.
>
>
> **Q5: Clarification on Qwen vs. DeepSeek**
>
> We agree that DeepSeek base models may be trained on chat data. However, the exact pretraining procedures remain undisclosed, limiting verifiability.
>
> Figure 3 (left) shows that Qwen can answer questions directly **without prompt templates**, implying it may have been pretrained on concatenated question-answer pairs directly. This supports your suggestion that Qwen may benefit from more **explicitly structured** training data.
>
>
> **Q6: Clarification regarding Section 3.3**
>
> We believe you intended to refer to **Section 3.3 (Line 244)** rather than Section 3.2. The cited sentence pertains to **Figure 6**, which evaluates both the effect of templates and training data coverage.
>
> The results show that training on ASDiv—a small (2K) dataset of basic algebra problems—yields significant performance gains. This supports our claim that “even a small and completely out-of-distribution question set could induce the reasoning ability equally well, by reinforcing useful reasoning behaviors rather than injecting new knowledge.”
>
>
> **Q7: Takeaway from LLaMA experiments**
>
> We agree that the initial underperformance of the red curve in Figure 7 (left) stems from suboptimal formatting. However, after RL training, this variant shows the most significant improvement and ultimately achieves the best performance among all runs.
>
> This result supports our main takeaway: **math-pretrained models benefit more from RL fine-tuning**.
>
> That said, we do **not** claim that NuminaMath is the optimal choice for pretraining. In our experiments, we simply chose NuminaMath and concatenated all available question–solution pairs from it for continual pretraining. This decision was motivated by our observation (Section 2.2) that Qwen models appear to have been pretrained on question–answer concatenations.
>
> ---
>
> Thank you once again for your time and insightful feedback!

---

> > ### Comment · Reviewer_FjK5 · 2025-06-07
> >
> > **Q1: response length bias**
> >
> > isn't this constant of proportionality the average due to an expectation?
> > wouldn't the expectation of a loss using each response length be the same as one using the average response length?
> >
> > **Q2: length bias**
> >
> > the results in the follow-up experiments seem to suggest it isn't the length normalization but the std deviation normalization that is causing this, no?
> >
> > **Q4: performance gains**
> >
> > If we're not expecting performance gains (or only marginal ones), this quite empirical paper doesn't seem very convincing
> > If the incorrect responses are truly shorter, I think you can make a reasonable argument for training efficiency since GRPO's incorrect answers during training will take longer to generate. Do you see any speed gains in your setup?
> >
> > **clarifications**
> >
> > **Q5: qwen vs deepseek pretraining**
> >
> > would be nice if there were more insight on this, but thanks for the clarification
> >
> > **Q6: section 3.3**
> >
> > I think the word "otherwise" is confusing here, because it reads like "otherwise (without templates)" where ASDiv does not improve performance.
> >
> > I think you should rephrase it to something like "Countering template mismatch can be done even with a very small and OOD dataset (e.g. ASDiv), which can regains reasoning ability for a new different template"
> >
> > **Q7: takeaway**
> >
> > I think this is close to an interesting takeaway but not fully justified
> >
> > math-pretrained models benefit more from RL fine-tuning *for math*
> >
> > perhaps math transfers to coding and other RLVR setups? but could it just be that having an initial baseline level of performance is necessary to improve performance with RL, regardless of domain? and perhaps there can also be latent knowledge of a domain within a model that is obscured by bad template and can confound tests of baseline accuracy.

---

> > > ### Author Response · Authors · 2025-06-08
> > >
> > > Thank you for your follow-up questions! We address them below:
> > >
> > > **Q1: (Theoretical) Length Bias**
> > >
> > > We respectfully disagree that the average arises from the expectation. Please refer to Appendix A of our paper for a detailed derivation of the policy gradient. We emphasize that the final policy gradient (Eq. 6) **does not** include any term related to response length. In contrast, the original GRPO formulation (with the 1/|o| factor) cannot be derived from the policy gradient and is therefore biased.
> > >
> > > **Q2: (Empirical) Length Bias**
> > >
> > > Our results indeed suggest that the **length normalization term** is responsible for the observed differences in both performance and output length instead of the std normalization term. For further clarification on naming, please refer to our response in another comment.
> > >
> > > **Q4: Performance & Efficiency**
> > >
> > > Figure B shows a performance gap between Dr. GRPO and GRPO. Additionally, we observe speed gains in our setup, as evidenced by the `actor/total_time` metric in the first row of Figure B.
> > >
> > > **Q6: Section 3.3**
> > >
> > > Thank you for your suggestion — we will carefully incorporate it in the next version of the paper.
> > >
> > > **Q7: Takeaway**
> > >
> > > Thank you for your thoughtful comments. We fully agree that "having an initial baseline level of performance is necessary to improve performance with RL, regardless of domain." This aligns with our claim in Section 2.1 regarding **R1-Zero Trainability: Templates Construct Exploratory Base Policies**. Specifically, we argue that as long as the base model (initial policy) achieves a sufficient level of performance (the base policy is exploratory), RLVR can further improve it.
> > >
> > > In this paper, we used the math domain as an example to study the R1-Zero-like training paradigm. We agree that extending this analysis to other RLVR domains would be an interesting direction for future work.
> > >
> > > ---
> > >
> > > Thank you again for your follow-up questions — they have helped us improve the clarity of the paper. We are happy to address any further concerns you may have.

---

> ### Author Response · Authors · 2025-06-04
> **Results for Q3 - ablation and statistical significance**
>
> Dear reviewer FjK5,
>
> Thank you again for your insightful suggestions. We’d like to follow up with the results of the additional experiments you proposed, focusing on ablation studies and statistical significance.
>
> ### Experimental Setup
> * **Model**: [Qwen/Qwen2.5-1.5B](https://huggingface.co/Qwen/Qwen2.5-1.5B).
> * **Dataset**: A mixture of 3K diverse math questions drawn from ASDiv, MATH, and AIME (pre-2023)
> * **Evaluation**: As in the paper, we report the average accuracy across five math benchmarks: MATH500, AMC, AIME24, Minerva Math, and OlympiadBench
> * **Hyperparameters**: All settings follow the code provided in the supplementary materials
> All other hyper-parameters follow the code in supplementary materials.
>
> ### Results
>
> **1: Ablation Study**
>
> [Figure A](https://postimg.cc/4Yzj3Qn2) presents the training and evaluation curves for the following variants:
> 1. Vanilla GRPO
> 2. GRPO w/o length normalization
> 3. GRPO w/o standard deviation (std) normalization
> 4. **Dr. GRPO** (our final method)
>
> From the first row of Figure A, we observe that both Dr. GRPO and the variant without length normalization generate shorter responses compared to the other two. This confirms that the length bias term has a more significant influence on response length—consistent with our expectations.
>
> In terms of performance, Dr. GRPO and the other ablated variants consistently outperform vanilla GRPO on both training rewards and evaluation accuracy. This indicates that removing bias terms (either length or std) improves policy learning, validating our motivation for Dr. GRPO.
>
>
>
> **Result 2: Statistical significance**
>
> [Figure B](https://postimg.cc/fkJFdNJ9) compares GRPO and Dr. GRPO across three independent runs. We observe that Dr. GRPO consistently demonstrates statistically significant improvements—both in token efficiency and final accuracy—across different random seeds.
>
> ---
>
> We appreciate your recommendation to include these analyses. These results strengthen our claims, and we will incorporate them into the next revision of the paper.

---

> > ### Comment · Reviewer_FjK5 · 2025-06-07
> >
> > Thank you for the ablation experiments. I have some followup questions and concerns
> >
> > **how many seeds are you using in the main paper?**
> >
> > the results here seem very noisy, do you only have this one follow-up experiment with 3 seeds?
> >
> > **isn't the no-length normalization run doing better than Dr. GRPO?**
> >
> > in Figure A, the no-len run is seemingly learning faster, achieving the same eval accuracy earlier, and it also has a comparable output length
> >
> > this seems to suggest that my theory about length normalization is accurate and that removing the division by std deviation is the major contribution here
> >
> > can you give an explanation of how both method changes interact and a takeaway?
> > perhaps an extra graph that would help is the eval response length separated by correct and incorrect answers

---

> > > ### Author Response · Authors · 2025-06-08
> > >
> > > Thank you for your followup questions!
> > >
> > > **Number of seed**
> > >
> > > Yes we used three seeds. The first row of Figure B looks noisy because it plots the training metrics. The second row of Figure B shows the evaluation results, which more clearly differentiate the two methods in terms of both performance and length.
> > >
> > > **Length normalization**
> > >
> > > Yes in Figure A, the `no-len` run is "learning faster, achieving the same eval accuracy earlier, and it also has a comparable output length". This run corresponds to `GRPO w/o length normalization` (hence the name `no-len`), and supports our claim regarding the length bias. It does not imply "removing the division by std deviation is the major contribution". We hope this clarifies the naming and intent.
> > >
> > > Thank you again for the thoughtful discussion — we are happy to address any further questions you may have.

---

### Decision · Program_Chairs · 2025-07-08

**Decision:**

Accept

**Comment:**

The paper studies the RLVR (reinforcement learning with verifiable rewards) setting, exploring the different base models and the details of the RL algorithm. The paper makes several observations:
- They study the effect of prompt template. For Qwen, no template performs best. For other models, appropriate template is needed.
- For all the base models they consider, they find "Aha moment" behaviors, i.e. self-correction and backtracking, in the samples from the base model, without RL.
- They show that there is a strong interaction between the observed improvement from RL, and the prompt template being used.
Overall, these results suggest that some of the observations made about RL may be misleading, and may be caused by base models which are already distilled / pretrained on reasoning-formatted data.

Finally, the authors observe issues with the standard GRPO method, which they argue could cause bias towards longer answers. First, they remove the normalization of advantages by the group reward std. Second, they remove the normalization by the sequence length, which is normally applied per-sequence.

This second modification has also been proposed in a concurrent work [1], and [2] also got rid of normalization by reward std (although they use a different way of estimating advantages). Both of these works are concurrent, so should not be viewed as reducing the novelty of this paper.

The authors show that the modified GRPO (dr. GRPO) achieves the same reward with less tokens per problem.

Overall, this paper makes several interesting observations and a reasonable methodological contribution. The main weakness in my opinion is the quality of presentation: the paper is fairly informal, and the structure is not very clear. Often, it is hard to know what exactly is meant e.g. by "Aha moment".

In the discussion period, the authors also did not respond very clearly to the question of reviewer FjK5 about the justification for per-token normalization. Overall, I believe the proposed modification is reasonably justified, and the experiments support the claims made by the authors.

Requested changes: Please include ablation of dr. GRPO pieces and comment on the statistical significance of the results (how many seeds use, how results are aggregated). Please respond to the theoretical question of reviewer FjK5.

[1] DAPO: An Open-Source LLM Reinforcement Learning System at Scale
[2] VAPO: Efficient and Reliable Reinforcement Learning for Advanced Reasoning Tasks